# Ontogenetic shifts in space use and habitat selection of tiger sharks (Galeocerdo cuvier) in The Bahamas

Kathryn A. Ayres ®*, Austin J. Gallagher

Beneath The Waves, Boston, Massachusetts, United States of America

* katy@beneaththewaves.org

## Abstract

Tiger sharks (Galeocerdo cuvier) display notable variation in behavior, movement patterns, and habitat use, which reflects ontogenetic shifts in diet and access to habitat types. In this study, we analyzed a comprehensive five-year dataset (2018–2023) of acoustically tagged tiger sharks (n = 39) across two island regions in The Bahamas. Network analysis revealed life-stage-specific differences in space use and habitat selection, with a significant correlation between motility (measured as edge density within arrays) and shark size (fork length, FL). Generalized Additive Models (GAMs) indicated that female sub-adult tiger sharks (225–250 cm FL) exhibited broader and more interconnected movements than juvenile and mature sharks, which were closely associated with shallow seagrass habitats. For male tiger sharks, fork length showed a positive linear relationship with edge density. We estimated fork length at the time of first detection for analysis, rather than measurements taken at tagging, and emphasize the importance of this approach when examining ontogenetic patterns in tiger sharks, which exhibit rapid growth during early life stages. Additionally, adult, and sub-adult sharks were recorded at receiver stations along the east coast of the U.S., highlighting large-scale partial migratory behavior and reiterating the need for transboundary conservation strategies.

## Introduction

Ontogenetic changes in habitat use and home-range size are common among vertebrate species, often driven by dietary shifts that correspond with increasing energy demands [1]. Typically, there is a positive correlation between body size and home-range [2,3]. These relationships have been widely documented among terrestrial vertebrates [4,5], however, there is comparatively less evidence for these patterns among large marine carnivores. In many large-bodied shark species, adults undertake extensive migratory movements and exhibiting site fidelity, often returning to specific locations seasonally or annually for reproduction or foraging purposes [6–8]. Juvenile sharks generally exhibit a preference for shallow, sheltered coastal areas before attaining sizes that

**Data availability statement:** Data are available via Zenodo at the following DOI: https://doi.org/10.5281/zenodo.17372500.

**Funding:** This work was supported by grants to BTW from the following funders: The Wanderlust Fund, The King Family, Sternlicht Family Foundation, Lush, Maverick1000, National Geographic Wild, Seaworld and Busch Gardens Conservation Fund, Thayer Academy, WCPD Foundation, and Anomaly Entertainment. The funders did not play any role in the study design, data collection and analysis, decision to publish, or preparation of the manuscript.

**Competing interests:** The authors have declared that no competing interests exist.

enable offshore and more wide-ranging movements [9]. Emerging evidence suggests that sub-adult sharks may exhibit broader-scale movements than large mature adults as they explore to establish suitable home-ranges [10–13], challenging the assumption of a strictly positive linear relationship between body size and home-range.

The tiger shark (*Galeocerdo cuvier*) is a large (< 5 m in total length), globally distributed species that occupies both coastal and oceanic habitats [14]. They are generalist feeders with a highly adaptable diet that shifts with age: juveniles primarily consume small prey such as teleosts, crustaceans, and reptiles, whereas adults feed on larger prey, including turtles, birds, other elasmobranchs, and marine mammals [15–17]. This dietary and ecological flexibility is reflected by substantial intraspecific variation in movement and behavior [18]. As apex predators, they exert strong top-down effects on marine food webs across diverse habitats, playing a critical role in structuring communities [19]. Despite their ecological significance, populations are declining worldwide, and the species is currently listed as *Near Threatened* on the IUCN Red List [20]. Understanding ontogenetic differences and identifying parturition areas is therefore essential for conservation planning [21], and while many coastal carcharhinids have well-documented, site-specific nursery grounds, these habitats remain poorly characterized for tiger sharks [22].

Investigating ontogenetic shifts in space use of tiger sharks can be challenging due to their highly mobile nature, but the use of satellite telemetry has shown that as they grow, they expand their vertical habitat use into deeper, more pelagic waters [9,23], undertake large-scale movements [24], and even complete transoceanic migrations [25,26]. Numerous studies have also documented connectivity between The Bahamas and the east coast of the U.S. [27–29], where broad areas of the continental shelf have been proposed as potential parturition habitats [30]. Acoustic telemetry has further revealed life-stage differences in residency at Bimini, The Bahamas [27], and long-term movements patterns in Hawaii [12]. In our previous work, we applied network analysis to acoustic tracking data from tiger sharks (*n* = 12) over a two-year period in The Bahamas [31] but limited sample sizes across life-stages precluded robust examination of ontogenetic patterns.

In the present study, we analysed a full five-year data set of a larger sample of tiger sharks over a more expansive acoustic array, which allowed us to compare movement and habitat selection across life-stages. Additionally, we determined large scale movements of tiger sharks linking The Bahamas with the eastcoast of the U.S. by leveraging a collaborative tracking database (Ocean Tracking Network). We hypothesize that tiger shark movement patterns, habitat use, and connectivity will differ across life-stages and expect sub-adult individuals to make the widestranging movements. Taken together, our results shed new light on the changing patterns of space use for this ecologically important species, illuminating divergent patterns between its early years of life through maturity.

## Materials and methods

### Study area

The Bahamas is an expansive subtropical archipelago, that supports diverse shark communities and has been a nationwide shark sanctuary since 2011 [32]. New

Providence (208 km²), is an island located centrally in the archipelago and its northern and western coasts feature sloping reef walls descending into the Tongue of the Ocean, while the southern and eastern regions are dominated by shallow seagrass and sandy habitats with patch reefs, forming the northern Great Bahama Bank. Great Exuma (158 km²), part of the Exuma Cays chain (roughly 230 km south of New Providence), lies along a reef tract separating the deep Exuma Sound to the east from shallow sand and seagrass meadows of the Great Bahama Bank to the west.

Acoustic receivers (Vemco VR2W, Vemco Ltd., Innovasea, Halifax, Canada) (n = 32) were deployed between February 2018 and July 2022 (S1 Table) in two arrays: the coastal waters around New Providence (n = 18) and Great Exuma (n = 14) (Fig 1). Six of these were deep receivers (VR2-AR) which were deployed using a sacrificial weight attached via a threaded release plug at the base of each unit. A high-pressure buoy maintained the receiver in an upright position within the water column. Maintenance and retrieval were carried out using an acoustic release command from a VR100 hydrophone (Vemco Ltd., Innovasea, Halifax, Canada). Acoustic receiver stations were classified depending on habitat type: coral reef (n = 9), deep wall (n = 6), sand (n = 3) and seagrass (n = 14) (S1 Fig). The depth of receivers ranged between 2 and 207 m.

Tiger sharks (n = 46) were tagged between 6th February 2018 and 30th May 2022 in New Providence (n = 16) and Great Exuma (n = 23) (Fig 1). Sharks were captured and tagged via drumline fishing, and, once boatside, were placed in tonic immobility and implanted internally with acoustic transmitters (V16, Innovasea) [31]. Sharks were also were measured and classified into the following life stages; young of the year (YOY < 110 cm FL), juvenile (< 180 cm FL), sub-adult (>180 cm), and adult (> 255 cm FL males, > 265 cm FL females) (males if claspers were calcified), based on a similar classification used previously in the region [27]. Total handling time ranged between 4 and 10 minutes.

### Ethics statement

The study was approved by the Canadian Council on Animal Care under administration of the Carleton University Animal Care Committee. There were no methods of sacrifice as the tagging procedures were non-lethal. To alleviate suffering sharks were continuously irrigated with seawater over the gills during handling to maintain oxygenation and were restrained for the shortest time possible.

### Data analysis

Detections were first filtered to remove potential false detections which were classified as detections that occurred within less than the minimum tag transmission delay or an individual detection. Filtered detections were then reduced to detection days (dates in which sharks were detected) to determine overall residency index (RI) which was calculated for each shark by dividing the total number of detection days by the monitoring period (time between the first and last detection). This provides the proportion of time that tiger sharks were detected within the acoustic arrays. The first and last detection of each shark was compared to the install and removal dates of each acoustic receiver to determine which stations in the arrays were active during the study period of each shark.

Network graphs were created using R package *igraph* with the acoustic receivers as 'nodes' and the movements (subsequent pings between receiver locations) as 'edges'. Occupancy, station residency, connectivity and betweenness were calculated for each station and for each individual and averages were compared across life-stage. Occupancy represents the average total number of detection days, station residency represents the number of consecutive detection days at a station before a) being detected at another station or b) being absent for over 24 hours and re-detected at the same station, connectivity represents the centrality degree (the number of connections with other stations) and betweenness measures how often a station lies on the shortest path between pairs of other stations in a network, reflecting how frequently it acts as a connector or bridge.

Node and edge densities were calculated for each individual shark for each of the two arrays, which represents motility as a proportion between 0 and 1. For node density, a value of 0 displays that a shark was detected at 0 receiver stations

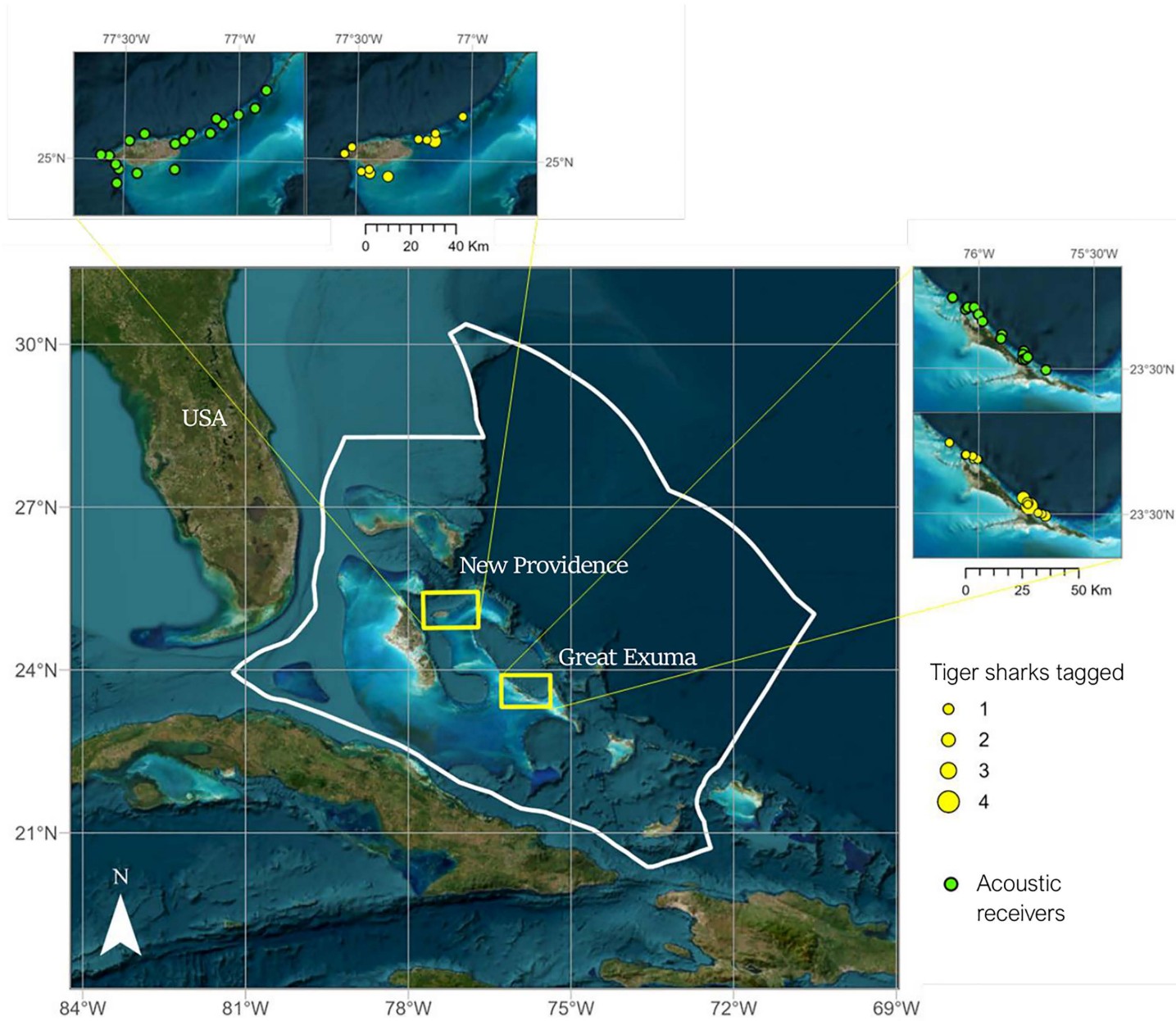

**Fig 1. Map of acoustic receiver stations and tagging locations of tiger sharks.** Map created in ARC GIS. Source of world map layer: Esri, Maxar, Earthstar Geographics, and the GIS User Community, "World Imagery" (basemap) Scale not given. February 19th 2012 https://www.arcgis.com/home/item.html?id=10df2279f9684e4a9f6a7f08febac2a9 (Accessed 7th January 2025).

and a value of 1 displays they were detected at all stations. For edge density, a value of 0 displays a shark that made 0 movements and a value of 1 displays they made every combination of movements throughout an array. Generalized Additive Models (GAMs) were built using the package '*mgcv*' [33] and used to evaluate relationships between fork length (FL cm), sex, island and node and edge densities, as these models allow for investigating relationships that are non-linear. Acoustic detections of tagged sharks were also checked on the Ocean Tracking Network (OTN) database to determine if sharks had been detected on other receivers installed in the broader region.

Sharks were first classified into life-stages based on fork length (cm) when initially tagged. For each individual, the elapsed time between tagging and first detection date was calculated, which ranged between 0-415 days (mean ± SE: 50 ± 15). Fork length at first detection was then estimated using growth rates between 4-40 cm FL per year [34,35] (S2 Table) and calculated at last detection. Five sharks that were initially classified as sub-adults were estimated to have reached maturity by the time of detection and were therefore reclassified as adults for analysis (S3 Table).

## Results

Tiger sharks (n = 39) were detected and estimated fork length at detection ranged between 91 and 300 cm FL (216 ± 8, mean ± SE). Males (n = 9) ranged between 140 and 253 cm FL (209 ± 15) and females (n = 30) between 91 and 300 cm FL (218 ± 9). Mean fork length did not differ significantly between sexes (two-sample t-test: $t = 0.48$, $df = 37$, $p > 0.05$; S2 Fig). Tiger sharks were classified into the following life-stages: YOY (n = 1), juvenile (n = 6), sub-adult (n = 22), and adult (n = 10) (S3 Table). Since only one YOY shark was tagged and detected, it was grouped with data for juvenile sharks for analysis.

Sharks were monitored between 1 and 1556 days (571 ± 81) and overall residency index ranged from 0.01 and 0.68 (0.26 ± 0.03). Overall residency did not significantly differ across life-stages, juveniles (0.20 ± 0.04) sub-adults (0.29 ± 0.04) or adults (0.23 ± 0.04). Residency did not significantly differ across months or life-stage, however fewer detection days were observed in summer months (S3 Fig).

### Network analysis

Network metrics for each individual shark were all significantly different from the random networks generated, determined using a one-sample Wilcoxon signed rank test (p < 0.01) [36]. Overall, a total of 30 tiger sharks displayed movements (were detected at more than one station). In the New Providence array, there were a total of 1769 movements displayed by 16 tiger sharks, and in the Great Exuma array, 1111 movements were displayed by 14 tiger sharks. Nine sharks (one juvenile, five sub-adults and three adults) also demonstrated connective movements between both arrays. These sharks did not, however, make movements within both arrays; they were detected at several stations within one array and then detected at only one station in the other, so for analysis the two arrays were kept as separate networks.

### New Providence

In the New Providence array, juvenile tiger sharks (n = 6) exhibited the highest frequency of movements between two shallow seagrass stations (N8 and N10; Fig 2A). Station residency (consecutive days detected) was greatest at coral reef station N1, the southernmost station in the array (Fig 2A), while occupancy (average total days) peaked at a seagrass station located in a shallow bay (N8; Fig 2D). Juveniles displayed low and relatively uniform centrality values across stations (Fig 2G), whereas stations located near the geographical centre of the network showed the highest betweenness values (Fig 2J).

Sub-adult tiger sharks (n = 6) were detected, with five individuals registering movements, most frequently between coral reef stations N3 and N4 (Fig 2B). Occupancy was highest at coral reef station N3 and seagrass station N10 (Fig 2E). Deep wall station N11 displayed slightly higher centrality than other stations (Fig 2H) and exhibited the highest betweenness (Fig 2K).

Adult tiger sharks (n = 5) were detected, all of which made movements, most were concentrated between shallow seagrass stations N8, N10, N12, and N14 (Fig 2C). Station residency was generally low across all stations. Occupancy peaked at seagrass station N12 and its adjacent sites (Fig 2F). Centrality was also highest at seagrass station N12, which functioned as a hub for neighbouring stations (Fig 2I). Betweenness was greatest at seagrass station N13 and deep wall station N11 (Fig 2L).

### Great Exuma

In the Great Exuma array, three juvenile tiger sharks were detected exclusively at sand station E12, (Fig 3D). No movements were recorded (Fig 3A), resulting in centrality and betweenness values of zero across all stations (Figs 3G and 3J). Sub-adult tiger sharks were detected (n = 12), nine of which made movements. Most movements occurred between deep

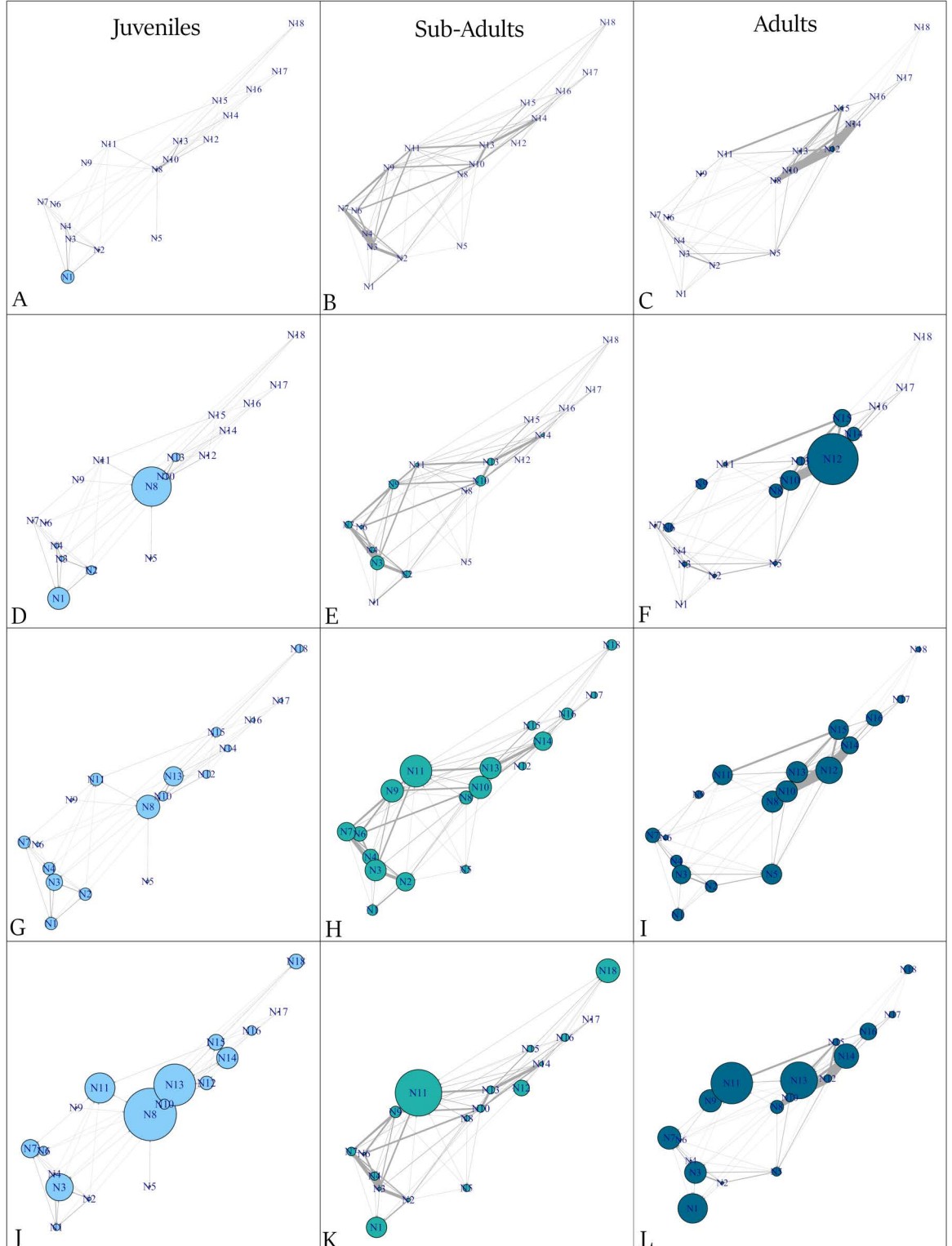

**Fig 2. Movement networks in New Providence, edge thickness represents number of movements for juvenile (n = 6), sub-adult (n = 6) and adult tiger sharks (n = 5) A-C) station residency (node size represents number of consecutive days detected) D-F) occupancy (node size representing average number of days detected) G-I) centrality index (node size represents the number of connections with other nodes) J-L) betweenness (node size representing how often a station lies on the shortest path between pairs of other stations in a network).**

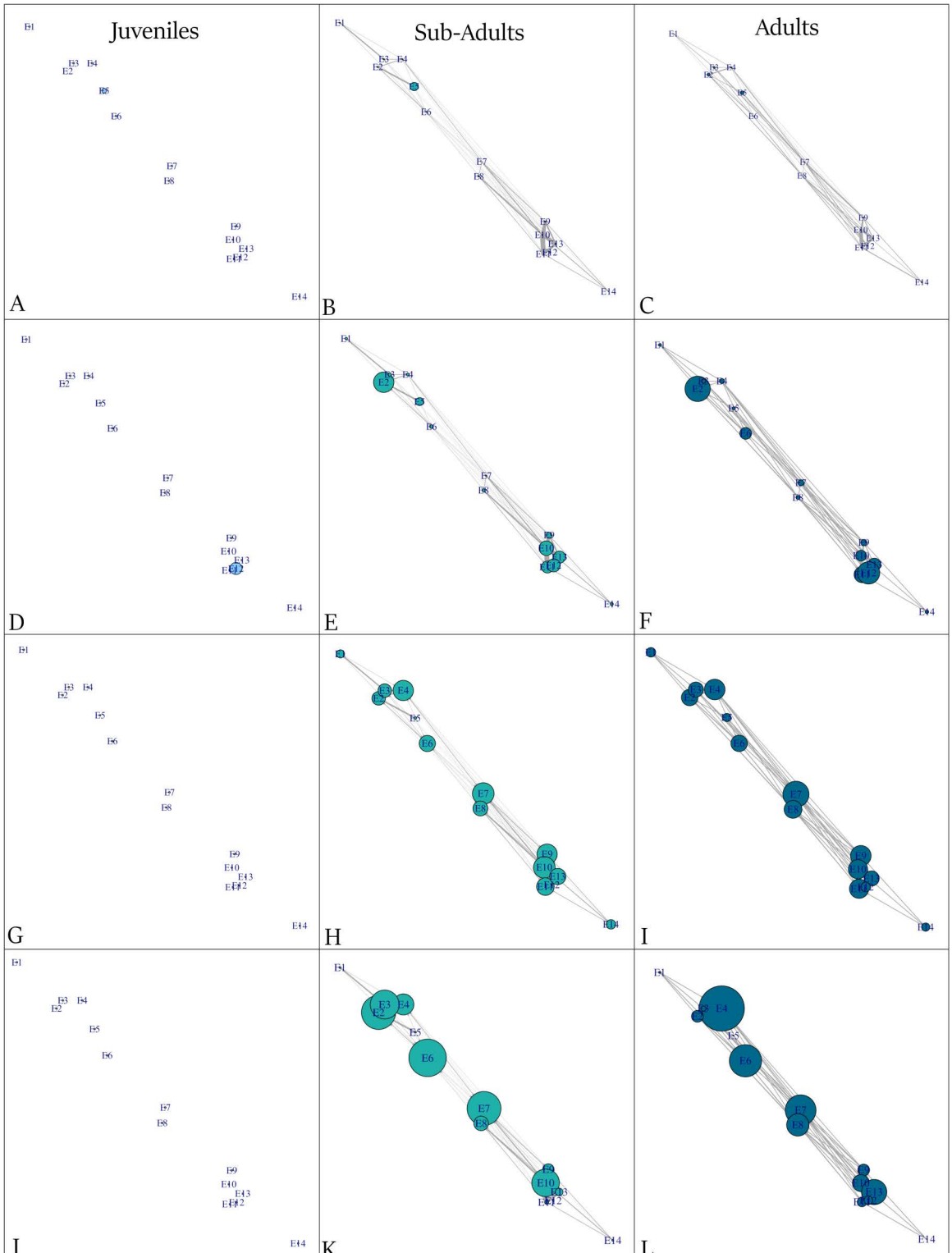

**Fig 3. Movement networks in Great Exuma, edge thickness represents number of movements for juvenile (n = 3), sub-adult (n = 12) and adult tiger sharks (n = 7) A-C) station residency (node size represents number of consecutive days detected) D–F) occupancy (node size representing average number of days detected) G-I) centrality index (node size represents the number of connections with other nodes) J-L) betweenness (node size representing how often a station lies on the shortest path between pairs of other stations in a network).**

wall station E9 and nearby seagrass stations E10, E11 and E13 (Fig 3E). Occupancy was highest at sand station E2 in the north and the seagrass stations in the south (Fig 3E). Centrality was generally high across stations, with the exception of coral reef station E5 and peripheral stations E1 (seagrass) and E14 (coral reef) (Fig 3H). Betweenness was lowest at the peripheral northern and southern stations (Fig 3K) Adult tiger sharks were detected (n=7), with five individuals registering movements. Network metrics were very similar to sub-adults, indicating no clear difference in movements patterns.

### Sexual differences in space-use

In the New Providence array, very few males were detected (n = 2) or registered movements (n = 1) (Fig 4B), and only three were initially tagged at this location (n=3). In contrast, females (n=15) were primarily detected between the seagrass stations (N8, N10, N13, N14) in the eastern sector of the array (Fig 4A), reflecting more restricted and localized movements. In Great Exuma, males (n=6) (Fig 4D) and females (n=8) (Fig 4C) displayed similar movement patterns throughout the array.

### Motility

Generalized Additive Models (GAMs) models were used to evaluate relationships between network analysis metrics: node density (ND), edge density (ED) and size (fork length, FL), sex, and island (array). Node density ranged between 0.13 and 1.00 (0.54±0.04, mean±SE) and had a positive relationship with fork length but this increase was not significant nor did it differ between sex or island (p>0.05). For edge density, model selection using AIC indicated that several models were similarly well-supported (ΔAIC<2) (Table 1). While model (ED~s(FL) + Island) had the lowest AIC, we selected the sex-specific smooth model (ED~s(FL, by = Sex) + Sex) as it provides greater biological interpretability with minimal loss of support and had the highest deviance explained (%).

Edge density ranged between 0 and 0.27 (0.11±0.01, mean±SE) and significantly increased with fork length in both sexes, but the relationship differed by sex: for males, edge density increased linearly with length (F=12.76, p=0.001), whereas for females, the effect was non-linear (F=2.81, p=0.039) (Fig 5 and the model predicted sub-adult tiger sharks (225–250 cm FL) to exhibit more motility.

### Regional movements

Seven female sharks, ranging between 176 and 273 cm FL (one juvenile, three sub-adults, three adults) were detected on acoustic receiver stations outside of The Bahamas, from separate research groups (within the OTN network) along the east coast of the U.S., across every state from Florida to Virginia. Four sharks were then re-detected back in our arrays in The Bahamas (Fig 6). One adult (ID_22692) was also detected at other arrays in The Bahamas, one station off Eleuthera (Freetown) and another off North Cat Cay before detection at stations outside of The Bahamas EEZ. The largest one directional movement was by an adult female shark (ID_60472) that travelled ~1500 km north to Virginia. Detections on the east coast of the U.S. were most common during the summer months (June to September) which coincides with the lower number of detection days during summer in our arrays in The Bahamas (S1 Fig).

### Discussion

Our results demonstrate differences in tiger shark habitat selection and movement patterns across ontogeny, throughout an expansive sampling region of The Bahamas and the surrounding region of the subtropical Atlantic. The use of network analysis facilitated the rapid and intuitive visualization of spatial use patterns and not only identified the most frequently visited sites but also highlighted the functional connectivity between habitat types. Our key findings include the comparatively wide-ranging movements of female sub-adults (medium-sized sharks), the identification of a hotspot area used by adult females, and evidence of a positive linear relationship between size and motility in male sharks. These differences may result due to shifts throughout growth and development, as habitats present trade-offs between foraging opportunities, predation risk, competition, and avoidance, that we discuss below.

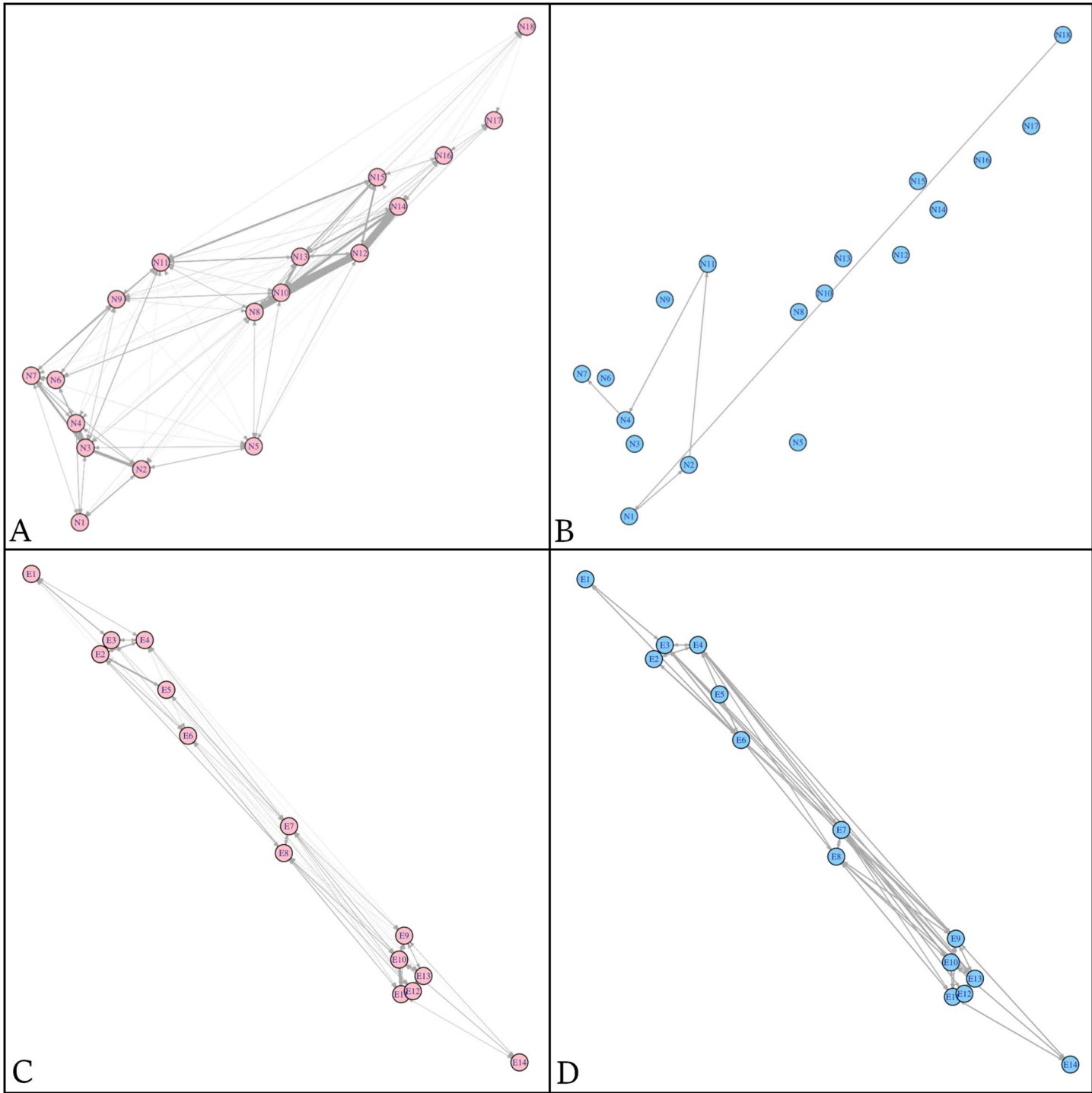

**Fig 4. Movement networks for tiger sharks, edge thickness represents number of movements A-B) New Providence, females (n = 15), and males (n = 1) C-D) Great Exuma, females (n = 8) and males (n = 6).**

**Table 1. GAM Models used to explore relationships between edge density and fork length (FL cm), sex, and island.**

| Model | AIC | ΔAIC | Weight | Deviance Explained (%) |
|---|---|---|---|---|
| ED~s(FL) + Island | −80.99 | 0.00 | 0.23 | 46.40 |
| ED~s(FL) | −80.86 | 0.13 | 0.21 | 42.40 |
| **ED~s(FL, by = Sex) + Sex + Island** | **−80.17** | **0.82** | **0.15** | **50.50** |
| ED~s(FL, by = Sex) + Island | −79.73 | 1.26 | 0.12 | 46.30 |
| ED~s(FL) + Sex + Island | −79.73 | 1.26 | 0.12 | 47.40 |
| ED~s(FL, by = Sex) + Sex | −79.03 | 1.96 | 0.09 | 45.20 |
| ED~s(FL) + Sex | −78.96 | 2.02 | 0.08 | 42.30 |
| ED~Sex | −67.83 | 13.16 | 0.00 | 0.53 |
| ED~Island | −67.76 | 13.23 | 0.00 | 0.32 |

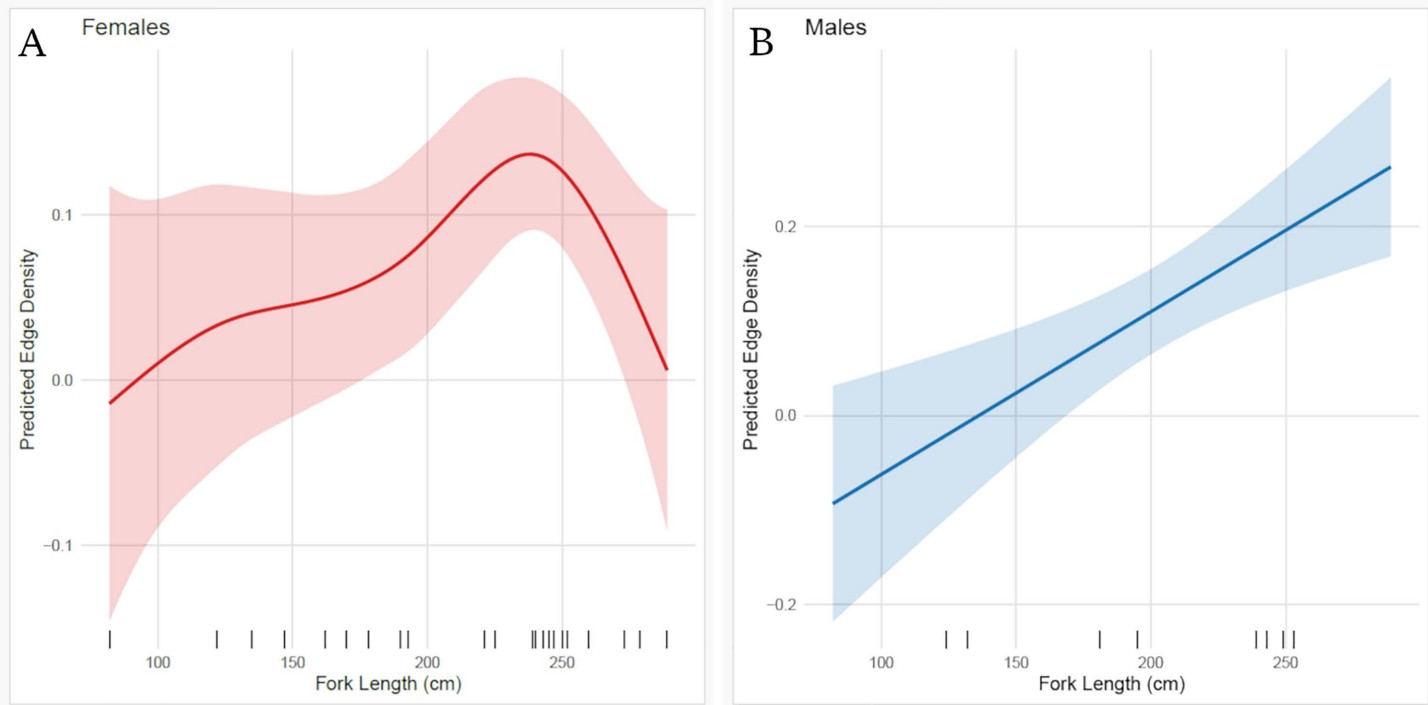

**Fig 5. Tiger shark motility within the networks at New Providence and Great Exuma, significant relationship between edge density and fork length (cm) A) females B) males.**

Juvenile tiger sharks had the highest occupancy in two station locations: the southern coral reef sector of the New Providence array and at a shallow, protected seagrass meadow (inside a bay) in the eastern sector. The southern station is situated on a fringing reef adjacent to a deep-water drop-off, an environment that likely provides foraging opportunities for juvenile tiger sharks on reef-associated teleost fish. This interpretation is consistent with findings from the northwest Atlantic where stomach content analyses revealed that teleost fish represent the predominant functional prey group for juvenile tiger sharks [39]. The proximity to the deep-water drop-off (< 1000 m) could also represent a strategy to reduce exposure to predation from larger conspecifics, with deeper water enhancing manoeuvrability and thus reducing the likelihood of capture [40]. In contrast in seagrass habitats, juveniles may benefit from enhanced camouflage due to their

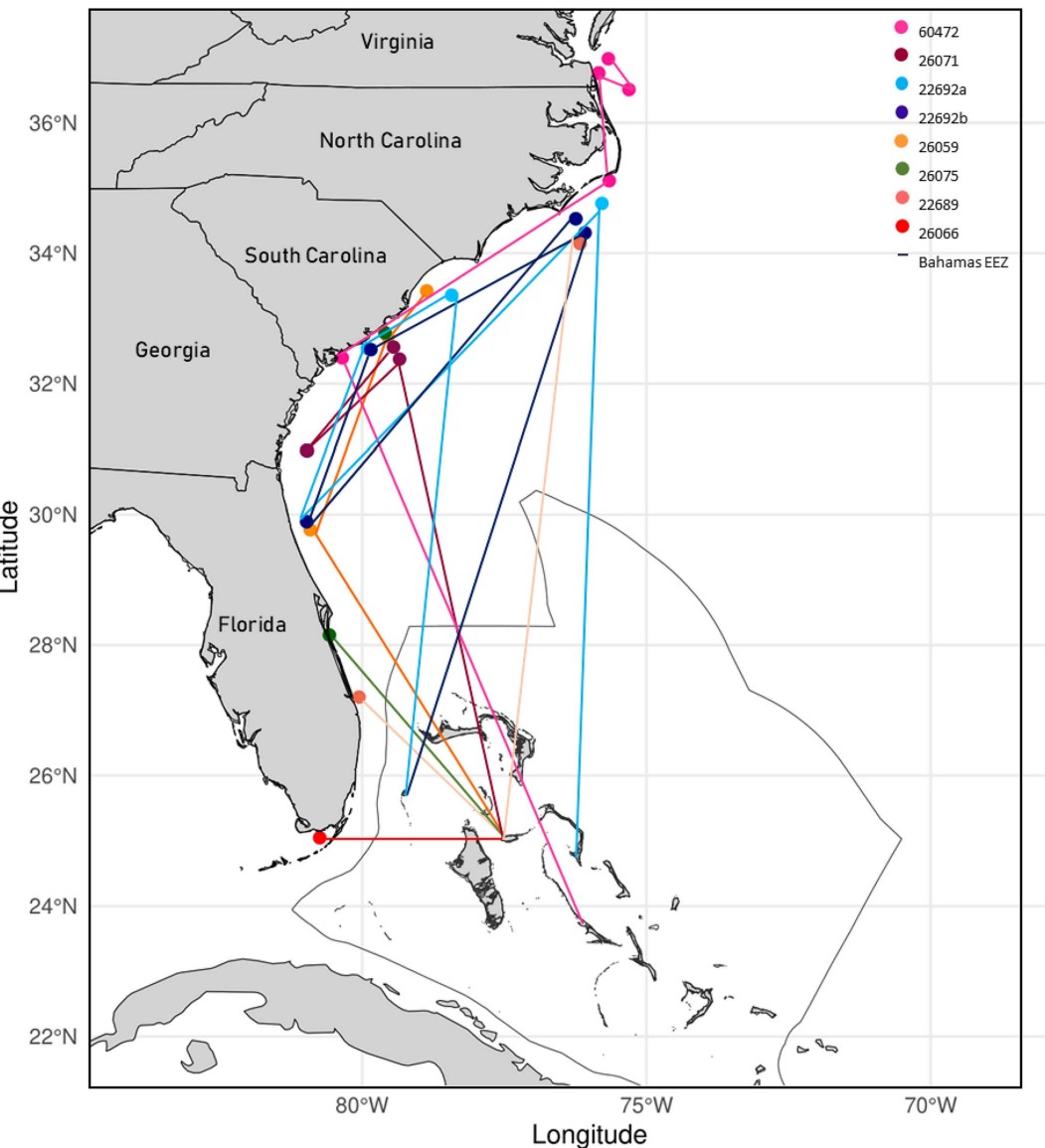

**Fig 6. Large scale connective movements of female tiger sharks acoustically tagged in The Bahamas, (tiger shark with ID_22692 migrated on two separate years a/b).** The map was generated in R studio using the rnaturalearth [37] and ggplot2 [38] packages with vector data from Natural Earth (version 4.1.0, 2023).

distinctively striped body pattern, which could reduce the risk of predation. They may also exploit these sheltered bays for foraging, targeting slow-moving benthic prey such as wrasses, crustaceans, and stingrays [41]. In addition, juveniles could be selecting the warm, shallow waters of the bay to enhance growth rate, thereby reaching a sufficient size for large-scale movements. Despite extensive sampling over approximately five years, neonate tiger sharks were not captured and only one individual was classified as YOY. We therefore propose that the waters surrounding New Providence may function as a secondary nursery area, but do not serve as a primary parturition site.

Sub-adult tiger sharks made expansive movements throughout both arrays and displayed no specific areas of high station residency, occupancy, or pattern in habitat preference. For females, sub-adult sharks had the highest edge densities

(used more possible edges in the networks) and the GAM model predicted sharks between 225 cm and 250 cm FL to exhibit more movement combinations than the smallest and largest sharks, supporting our hypothesis. This pattern may reflect an expansion of search behavior as sharks grow and encounter a broader diversity of prey. The continuous movement could also allow sharks to ambush unsuspecting prey and warrants more chance of a successful predation than remaining within the same area [41]. This was likely the case for all detected sharks as station residency was low across all stations (number of consecutive days detected at a station). In comparison, larger sharks exhibited more focused and restricted movements, having acquired the experience and capability to efficiently capture prey [42]. The inverse relationship between space-use and body size between larger and mid-sized sharks has also been demonstrated for acoustically tagged tiger sharks in Hawaii [12] and other large species including bluntnose six-gill sharks (*Hexanchus griseus*) [10] and white sharks (*Carcharodon carcharias*) [11,13].

Large female sharks displayed highly localized movements among four neighboring seagrass-stations on the eastern sector of the array in New Providence. This seagrass habitat may represent a hotspot for sea turtles, a predominant seagrass grazer throughout The Bahamas, potentially explaining the elevated shark activity and spatial use observed in the area. A future study is warranted to investigate this ecological linkage. In the Galapagos, large tiger sharks exhibited the highest concentrations of movements in front of major turtle nesting beaches [43]. By contrast, in Australia, tracked tiger sharks remained at turtle beaches even outside of the nesting season, suggesting reliance on alternative food sources [44]. As forage-driven movements in tiger sharks are influenced by size, both large males and females might be expected to utilize the seagrass area hotspot in our study; however, only females were detected. Stable isotope analysis has revealed isotopic overlap between sexes of tiger sharks in The Galapagos, however results suggested that males exhibit a greater diversity of foraging strategies [17]. Consistent with this, the largest males in our study were predicted by our model to have the widest-ranging movements, potentially reflecting a similarly broader dietary diversity. The warm shallow seagrass area may also serve as a site for mature females to facilitate rapid growth of developing embryos or to refuge from male harassment. This behaviour has also been suggested at a shallow sand bank area known as 'Tiger Beach' (26.86°N, 79.04°W) located off Grand Bahama (approx. 250 km north of New Providence) [45].

Although sharks were tracked across life stages using the same receiver array, it is important to consider that habitat-specific differences in detectability could influence observed patterns. For instance, dense seagrass may attenuate acoustic signals, potentially underestimating connectivity for sharks frequenting these areas, whereas open reef or shelf habitats may allow for more reliable detection. Regarding array design, receivers were distributed across several habitat types; however, 14 of the 32 stations were located within seagrass due to its dominance in the study area. Deep sites were also limited due to the logistical challenges of installing receivers beyond the safe diving limits of SCUBA. These differences in detection probability could partially contribute to the observed life-stage-specific patterns of movement and space use. Additionally, not all receiver stations were active throughout the entire study period, a common logistical constraint in long-term acoustic telemetry studies, therefore, edge and node densities were calculated by only including stations that were installed and recording between each shark's first and last detection. While our analyses provide strong evidence for ontogenetic differences, these limitations should be considered when interpreting results.

Our study adds to the body of knowledge that tiger sharks display connective movements between The Bahamas and the east coast of the U.S., which has been documented previously using satellite telemetry [26,28], acoustic telemetry and mark recapture [27,46]. Indeed, mixed size classes of tiger sharks have been documented in these areas and show molecular connections to The Bahamas [47]. The sharks that made the movements in our current study were classified using fork length at first detection (one juvenile, three sub-adults and three adults), but the small juvenile (176 cm FL, ID: 26075) did not make this movement until June 2021, almost two years after being first detected in The Bahamas. Based on growth rate estimates (S3 Table) this shark could have been approximately 211 cm in fork length at the time of

migration and therefore a sub-adult. A key limitation of multi-year movement studies is that individual sharks continue to grow throughout the study period, introducing a confounding factor that becomes more pronounced with longer datasets. Consequently, size measurements taken at the time of tagging may not accurately represent the body size of sharks when subsequent movements are recorded. Similar patterns were observed in tiger sharks tagged off Bermuda, in which two small individuals exhibited migratory movements during their second year of monitoring, but by at which time, based on growth rate estimates, had reached a sufficient size to do so [48]. We recommend that ontogenetic studies account for this effect by considering the elapsed time between tagging and first detection, which for several sharks in our study exceeded one year.

The low overall residency of detected tiger sharks (0.26) indicates that individuals spent approximately 74% of their monitoring period outside the two acoustic arrays. Given the extensive Bahamian EEZ and detections at other local acoustic arrays, this does not necessarily imply that sharks were outside of the Bahamian shark sanctuary and exposed to fishing. Global ocean warming has however been shown to strongly influence the migrations of highly mobile species closely associated with sea surface temperature [49], and long-term modelling of tiger sharks has revealed a poleward shift in their distributions in response to rising ocean temperatures [50]. These shifts have important implications for MPAs at lower latitudes, such as The Bahamas. Despite these findings, tiger shark populations in The Bahamas currently appear stable or slightly increasing [51], but it is important that this is monitored.

In conclusion, our study highlights clear and significant ontogenetic shifts in space use, habitat selection, and movement patterns of tiger sharks in The Bahamas, a region which we believe confidently supports the most diverse and abundant strongholds of this critically-important marine species. Over three decades of published research into the biology and behaviour of tiger sharks in this large ocean island state supports this notion, with the body of work documenting small-scale residency, partial and full migration, and philopatry to The Bahamas. A detailed network analysis into the data provided by these apex predators, as performed here, supports, and advances previous work, ushering new perspectives and questions for a species which is significantly more complex, and likely more important, than previously considered.

## Supporting information

**S1 Fig.  Habitat type and location of acoustic receivers in New Providence and Great Exuma, The Bahamas.**
(DOCX)

**S2 Fig.  Estimated fork length (cm) size at detection for female (n = 30) and male tiger sharks (n = 9).**
(DOCX)

**S3 Fig.  Average number of detection days of acoustically tagged tiger sharks in each month at New Providence and Great Exuma, The Bahamas and season (warm/cool).**
(DOCX)

**S1 Table.  Acoustic receiver station names, island, depth, habitat, code names and locations.**
(DOCX)

**S2 Table.  Growth rate estimates per shark size class based on Branstetter et al. 1987.** Table modified from Wirsing et al. 2006.
(DOCX)

**S3 Table.  Attributes of tiger sharks that were detected on the acoustic receiver arrays.**
(DOCX)

 

## Acknowledgments

We thank our Bahamian partners and stakeholders who have enabled and supported this work: L. Gittens and the Dept. Marine Resources, E. Carey and S. Cant from Bahamas National Trust, S. Cove from Stuart Cove's, J. Todd, P. Nicholson, R. Sands from Grand Isle Resort, A. Phillips and A. Musgrove from Bahamas Dive Guides, Dive Exuma, and The Exuma Foundation. We are particularly grateful to the following partners for logistical and operational support: Stuart Cove's, The International Seakeepers Association, The Grand Isle Resort, GIR Bahamas, M/YMarcato, J. and M. McClurg, Fleet Miami, R/V Sharkwater, Atlantis, Bahamas Dive Guides, Dive Exuma, The Exuma Foundation, Vemco, and the Ocean Tracking Network. For field work support, we thank B. Shea, O. Dixon, S. Aldridge, J. Halvorsen, E. Staaterman, E. Quintero, J. Sternlicht, T. Gilbert, J. Roth, S. Moorhead, E. Sudal, and M. Adunni. This work was covered under a permit to Austin Gallagher from the Department of Marine Resources.

## Author contributions

**Conceptualization:** Austin J. Gallagher.

**Data curation:** Kathryn A. Ayres, Austin J. Gallagher.

**Formal analysis:** Kathryn A. Ayres.

**Funding acquisition:** Austin J. Gallagher.

**Investigation:** Kathryn A. Ayres, Austin J. Gallagher.

**Methodology:** Kathryn A. Ayres, Austin J. Gallagher.

**Resources:** Austin J. Gallagher.

**Software:** Kathryn A. Ayres.

**Supervision:** Austin J. Gallagher.

**Writing – original draft:** Kathryn A. Ayres.

**Writing – review & editing:** Kathryn A. Ayres, Austin J. Gallagher.

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
