## [Decision Letter · Decision Letter 0]

15 Jul 2025

Dear Dr. Ayres,

Thank you for submitting your manuscript to PLOS ONE. After careful consideration, we feel that it has merit but does not fully meet PLOS ONE’s publication criteria as it currently stands. Therefore, we invite you to submit a revised version of the manuscript that addresses the points raised during the review process.

We look forward to receiving your revised manuscript.

Kind regards,

Joel Harrison Gayford

Academic Editor

PLOS ONE

Journal Requirements:

We thank our Bahamian partners and stakeholders who have enabled and supported this work: Gittens and the Dept.Marine Resources, E. Carey and S. Cant from Bahamas National Trust, S.Cove from Stuart Cove’s, J. Todd, P. Nicholson, R. Sands from Grand Isle Resort, A. Phillips and A. Musgrove from Bahamas Dive Guides, Dive Exuma, and The Exuma Foundation. We are particularly grateful to the following partners for logistical and operational support: Stuart Cove’s, The International Seakeepers Association, The Grand Isle Resort, GIR Bahamas, M/YMarcato, J. and M. McClurg, Fleet Miami, R/V Sharkwater, Atlantis, Bahamas Dive Guides, Dive Exuma, The Exuma Foundation, Vemco, and the Ocean Tracking Network. For field work support, we thank B. Shea, O. Dixon, S. Aldridge, J. Halvorsen, E. Staaterman, E. Quintero, J. Sternlicht, T. Gilbert, J. Roth, S. Moorhead, E. Sudal, and M. Adunni. This work was covered under a permit to Austin Gallagher from the Department of Marine Resources. We also thank our funders: The Wanderlust Fund, The King Family, Sternlicht Family Foundation, Lush, Maverick1000, National Geographic Wild, Seaworld and Busch Gardens Conservation Fund, Thayer Academy, WCPD Foundation, and Anomaly Entertainment.

This work was supported by grants to BTW from the following funders: The Wanderlust Fund, The King Family, Sternlicht Family Foundation, Lush, Maverick1000, National Geographic Wild, Seaworld and Busch Gardens Conservation Fund, Thayer Academy, WCPD Foundation, and Anomaly Entertainment. The funders did not play any role in the study design, data collection and analysis, decision to publish, or preparation of the manuscript

5. In the online submission form, you indicated that the raw data supporting the conclusions of this article will be made available by the authors upon request.

7. We note that Figures 1 and 7 in your submission contain map/satellite images which may be copyrighted. All PLOS content is published under the Creative Commons Attribution License (CC BY 4.0), which means that the manuscript, images, and Supporting Information files will be freely available online, and any third party is permitted to access, download, copy, distribute, and use these materials in any way, even commercially, with proper attribution. For these reasons, we cannot publish previously copyrighted maps or satellite images created using proprietary data, such as Google software (Google Maps, Street View, and Earth). For more information, see our copyright guidelines: http://journals.plos.org/plosone/s/licenses-and-copyright.

a. You may seek permission from the original copyright holder of Figures 1 and 7 to publish the content specifically under the CC BY 4.0 license. 

Reviewers' comments:

Reviewer's Responses to Questions

**Comments to the Author**

1. Is the manuscript technically sound, and do the data support the conclusions?

Reviewer #1: Partly

Reviewer #2: Yes

2. Has the statistical analysis been performed appropriately and rigorously?

Reviewer #1: Yes

Reviewer #2: Yes

3. Have the authors made all data underlying the findings in their manuscript fully available?

Reviewer #1: Yes

Reviewer #2: Yes

4. Is the manuscript presented in an intelligible fashion and written in standard English?

Reviewer #1: No

Reviewer #2: Yes

Reviewer #1: This manuscript titled "Space use and habitat selection of tiger sharks (Galeocerdo cuvier) across ontogeny throughout The Bahamas" (PONE-D-25-14869) presents an analysis of tiger shark spatial ecology using a five-year acoustic telemetry dataset from The Bahamas. The study is in a good position to advance our understanding of ontogenetic shifts in habitat use and regional connectivity of an ecologically important apex predator in a vulnerable subtropical ecosystem. The combination of network analysis, GAM modeling, and integration with the Ocean Tracking Network makes this potentially a strong and highly relevant contribution. However, improvements in organization, writing clarity, better contextualizing of the literature in the Introduction and Discussion would significantly strengthen the manuscript. Addressing the flow of the introduction and trimming repetitive results would greatly improve readability. A more critical reflection on methodology limitations (especially relating to array coverage and tag detection variability) is also recommended.

Major Comments:

- While the introduction contains relevant background information and appropriate citations, it suffers from issues with structure and flow. Several paragraphs appear fragmented or incomplete, and the transitions between key themes (such as ontogeny, conservation, and nursery habitat use) feel abrupt. The authors should consider reorganizing the introduction into a more streamlined narrative, potentially by combining certain paragraphs and ensuring each one clearly builds on the previous in a flowing manner.

- Additionally, the introduction frequently uses phrases like “has been documented” without clearly elaborating on the nature or significance of those findings. Rather than simply stating that movement patterns or nursery use have been previously reported, the authors should use this space to immediately and briefly explain how they were documented (e.g., via satellite telemetry, mark-recapture, etc.) and highlight what implications or gaps remain.

- A clearer comparison with existing literature (e.g., Gallagher et al. 2021, Hammerschlag et al. 2022) is needed to inform the reader in more depth and emphasize what is new in this paper beyond sample size.

- The final paragraph of the Introduction lacks the structure typical of hypothesis-driven studies. It transitions too quickly into a summary of the methods and findings, rather than clearly articulating the study's research objectives and hypotheses. Currently the only value of this study beyond others that I can see is the larger sample size, which typically does not warrant an additional study.

- The results section is very long and at times too descriptive. Consider streamlining if possible.

- The discussion is thoughtful and highlights ecological mechanisms, but it would benefit from stronger synthesis and focus on key findings to drive the value of this present study.

- The finding that sub-adults had highest edge densities is intriguing and consistent with exploratory behaviors during this life-stage. However, the inverse relationship with adults requires more discussion, especially in light of potential detection biases (e.g., seagrass areas with many receivers may inflate local detections).

- The GAMs seem appropriate but need further clarity; how were variables selected, and were interactions tested?

- The issue of shark growth over the study period is acknowledged, but the way this was addressed seems vague. Did the reclassification of life stages affect the statistical models?

- There are numerous typos, grammatical issues, and awkward phrasings (e.g., “and connectivity was low”, “most were between...”). A thorough review is needed before publication.

Minor Comments and Edits:

- Title: Consider rephrasing to make it more concise and active, e.g., “Ontogenetic Shifts in Space Use and Habitat Selection of Tiger Sharks in The Bahamas”

- It appears the figure captions are duplicated in the text and at the end with the figures.

- The manuscript refers to “Figure 6A,” which implies the existence of additional subpanels (e.g., 6B, 6C), but no such panels are presented or described.

- Good to see animal welfare and permitting addressed. However, consider briefly summarizing how long handling typically lasted and tagging success rate.

- As mentioned in the text, The Bahamas is an expansive archipelago. More attention should be paid to what areas in The Bahamas are being studied and why.

- Include the sample size and key finding about sub-adults’ motility.

- Line 35-40: Too many citations in a row. Add context or split sentences.

- Line 60: Protected from what?

- Line 252: “Gravid implications” is likely a typo for “grave implications”.

- Line 275: “female tiger sharks are also known to aggregate at Tiger Beach” → include GPS range or region coordinates if possible.

- Wordy expressions should be revised for clarity. Examples:

- Line 20: “acoustically detected at receiver stations” → simplify to “detected at receiver stations”

- Line 263: "sharks could also be selecting warm shallow seagrass to accelerate growth" → "sharks may use warm shallow seagrass to enhance growth rates"

- Line 275: “high concentration of movements within the same four adjacent shallow seagrass stations” → “highly localized movements among four neighboring seagrass-stations”

Reviewer #2: Line 76 - What are your specific Aims of the study? You state what you did above, but not what you are trying to resolve/address by this research. This will give your Discussion more direction. Has a network analyses been done before on tiger sharks? What can it tell us that's novel? What are the challenges with this approach for a highly mobile species like this?

Line 92 - Were sharks also placed in tonic immobility? Was any anesthesia used?

Line 94 - lower case s for sub-adult

Line 128 - Table S2 needs to include an 'Island' column, so the reader can understand which array they were caught and tagged in.

Figure 2 - Suggest as it's not a significant result, move to supp material.

Line 160 - Check your use of 'Adult' vs 'Adults' throughout, currently inconsistent.

Line 164 - 14 not fourteen (usually <10 is spelt out - disregard if this is a journal formatting rule).

174 - Were any tagged sharks from one island detected in the other over the duration of the study? This wasn't clear. If so/not, warrants some points in the Discussion.

254 - These different habitat types would be good to be captured on a map for readers.

264 - Reference needed.

274 - Reference needed.

278 - From Table S2 I can only assume (until Island is included as a column) that only one mature male was tagged in the New Providence array? Therefore, it's a bit of a stretch to state that only large females were detected here if hardly any mature males were tagged? I think the language here can be softened considerably on these points. Same with using the term 'aggregation' - probably needs some qualification of how you are classifying an aggregation herein. As tiger sharks are semi-solitary, is an 'aggregation' different to what we perceive it to be in a more schooling elasmobranch species?

282 - 'Concentration', or regular detection of large individuals moving through this section of the array? Watch for word choice again here.

284 - Small ones can 'chase' them too - they may not be as successful though. Suggest slight re-word.

306 - These last two paragraphs are overly focused on prey/diet drivers of habitat use. While important, there is no deep discussion about the habitat types that may be apparent from your detection data by shark size/sex, other possible drivers of space use in your arrays (detections time of day, current strength/direction, tides, other?). Considerations of these ideas may reveal something more important than just prey drivers and will strengthen your Discussion. It may help if you clarify your project Aims clearer above, it will guide your Discussion points better here.

316-328 - Suggest you re-think the points in the paragraph here and their relevance, as I don't recall MPA's being an important part of this study (again, coming back to characterising your Aims/Research Questions for the study earlier should help here).

350 - What about outside of The Bahamas? What applicability does the approach you took in this study have to other locations across the world? How is that important for a species like this that connects various regions globally? The Discussion is very parochial in its current form.

References - check you ref list for issues like italics for species names etc.

Figure 1 - Needs more details. Most readers won't be from The Bahamas. What islands are each that are zoomed in on? Can they be labelled? Can you also include a larger map of where The Bahamas is in relation to the USA? This is important as you discuss connectivity to the USA.

Figure 2 - Move to Supp material

Overall, this paper has the potential to be a good contribution to the existing literature on tiger sharks once the aforementioned points have been addressed.

**Do you want your identity to be public for this peer review?** For information about this choice, including consent withdrawal, please see our Privacy Policy

Reviewer #1: No

Reviewer #2: **Yes: ** Bonnie J Holmes

---

## [Author Response · Author response to Decision Letter 1]

28 Aug 2025

PONE-D-25-14869

Space use and habitat selection of tiger sharks (Galeocerdo cuvier) across ontogeny throughout The Bahamas

PLOS ONE

Dear Dr. Ayres,

Thank you for submitting your manuscript to PLOS ONE. After careful consideration, we feel that it has merit but does not fully meet PLOS ONE’s publication criteria as it currently stands. Therefore, we invite you to submit a revised version of the manuscript that addresses the points raised during the review process.

Both reviewers saw merit in the study, and have provided sensible recommendations for how the quality of the manuscript might be improved. Please bear these comments in mind when revising the manuscript.

We look forward to receiving your revised manuscript.

Kind regards,

Joel Harrison Gayford

Academic Editor

PLOS ONE

Journal Requirements:

Response: Style changed to match requirements in pdf.

Response – funding removed from the manuscript

We thank our Bahamian partners and stakeholders who have enabled and supported this work: Gittens and the Dept.Marine Resources, E. Carey and S. Cant from Bahamas National Trust, S.Cove from Stuart Cove’s, J. Todd, P. Nicholson, R. Sands from Grand Isle Resort, A. Phillips and A. Musgrove from Bahamas Dive Guides, Dive Exuma, and The Exuma Foundation. We are particularly grateful to the following partners for logistical and operational support: Stuart Cove’s, The International Seakeepers Association, The Grand Isle Resort, GIR Bahamas, M/YMarcato, J. and M. McClurg, Fleet Miami, R/V Sharkwater, Atlantis, Bahamas Dive Guides, Dive Exuma, The Exuma Foundation, Vemco, and the Ocean Tracking Network. For field work support, we thank B. Shea, O. Dixon, S. Aldridge, J. Halvorsen, E. Staaterman, E. Quintero, J. Sternlicht, T. Gilbert, J. Roth, S. Moorhead, E. Sudal, and M. Adunni. This work was covered under a permit to Austin Gallagher from the Department of Marine Resources. We also thank our funders: The Wanderlust Fund, The King Family, Sternlicht Family Foundation, Lush, Maverick1000, National Geographic Wild, Seaworld and Busch Gardens Conservation Fund, Thayer Academy, WCPD Foundation, and Anomaly Entertainment.

This work was supported by grants to BTW from the following funders: The Wanderlust Fund, The King Family, Sternlicht Family Foundation, Lush, Maverick1000, National Geographic Wild, Seaworld and Busch Gardens Conservation Fund, Thayer Academy, WCPD Foundation, and Anomaly Entertainment. The funders did not play any role in the study design, data collection and analysis, decision to publish, or preparation of the manuscript

Response: Removed from manuscript

Response: This is the funding information: This work was supported by grants to BTW from the following funders: The Wanderlust Fund, The King Family, Sternlicht Family Foundation, Lush, Maverick1000, National Geographic Wild, Seaworld and Busch Gardens Conservation Fund, Thayer Academy, WCPD Foundation, and Anomaly Entertainment. These funders were not involved in the study design, collection, analysis, interpretation of data, the writing of this article or the decision to submit it for publication.

Response: Only in Method section

5. In the online submission form, you indicated that the raw data supporting the conclusions of this article will be made available by the authors upon request.

Response: Data will be made publicly available

Response: Entire data will be made freely accessible

7. We note that Figures 1 and 7 in your submission contain map/satellite images which may be copyrighted. All PLOS content is published under the Creative Commons Attribution License (CC BY 4.0), which means that the manuscript, images, and Supporting Information files will be freely available online, and any third party is permitted to access, download, copy, distribute, and use these materials in any way, even commercially, with proper attribution. For these reasons, we cannot publish previously copyrighted maps or satellite images created using proprietary data, such as Google software (Google Maps, Street View, and Earth). For more information, see our copyright guidelines: http://journals.plos.org/plosone/s/licenses-and-copyright.

a. You may seek permission from the original copyright holder of Figures 1 and 7 to publish the content specifically under the CC BY 4.0 license.

Response: Figure 1: We created the map in ArcGis. Source of world map layer: Esri, Maxar, Earthstar Geographics, and the GIS User Community, “World Imagery’ (basemap) Scale not given. February 19th 2012 https://www.arcgis.com/home/item.html?id=10df2279f9684e4a9f6a7f08febac2a9 (Accessed 7th January 2025).

Figure 7. The map was generated in R using the rnaturalearth (South, 2023) and ggplot2 (Wickham, 2016) packages with vector data from Natural Earth (version 4.1.0, 2023) - public domain.

Response: Captions and citations included.

Reviewers' comments:

Reviewer's Responses to Questions

Comments to the Author

1. Is the manuscript technically sound, and do the data support the conclusions?

Reviewer #1: Partly

Reviewer #2: Yes

2. Has the statistical analysis been performed appropriately and rigorously?

Reviewer #1: Yes

Reviewer #2: Yes

3. Have the authors made all data underlying the findings in their manuscript fully available?

Reviewer #1: Yes

Reviewer #2: Yes

4. Is the manuscript presented in an intelligible fashion and written in standard English?

Reviewer #1: No

Reviewer #2: Yes

5. Review Comments to the Author

Reviewer #1: This manuscript titled "Space use and habitat selection of tiger sharks (Galeocerdo cuvier) across ontogeny throughout The Bahamas" (PONE-D-25-14869) presents an analysis of tiger shark spatial ecology using a five-year acoustic telemetry dataset from The Bahamas. The study is in a good position to advance our understanding of ontogenetic shifts in habitat use and regional connectivity of an ecologically important apex predator in a vulnerable subtropical ecosystem. The combination of network analysis, GAM modeling, and integration with the Ocean Tracking Network makes this potentially a strong and highly relevant contribution. However, improvements in organization, writing clarity, better contextualizing of the literature in the Introduction and Discussion would significantly strengthen the manuscript. Addressing the flow of the introduction and trimming repetitive results would greatly improve readability. A more critical reflection on methodology limitations (especially relating to

---

## [Decision Letter · Decision Letter 1]

24 Sep 2025

Dear Dr. Ayres,

Thank you for submitting your manuscript to PLOS ONE. After careful consideration, we feel that it has merit but does not fully meet PLOS ONE’s publication criteria as it currently stands. Therefore, we invite you to submit a revised version of the manuscript that addresses the points raised during the review process.

We look forward to receiving your revised manuscript.

Kind regards,

Joel Harrison Gayford

Academic Editor

PLOS ONE

Journal Requirements:

Reviewers' comments:

Reviewer's Responses to Questions

**Comments to the Author**

Reviewer #1: (No Response)

2. Is the manuscript technically sound, and do the data support the conclusions?

Reviewer #1: Yes

3. Has the statistical analysis been performed appropriately and rigorously?

Reviewer #1: Yes

4. Have the authors made all data underlying the findings in their manuscript fully available?

Reviewer #1: Yes

5. Is the manuscript presented in an intelligible fashion and written in standard English?

Reviewer #1: Yes

Reviewer #1: The authors have made a commendable effort to address the majority of my comments, and the manuscript is much improved in terms of organization, clarity, and methodological detail. However, I feel a few points remain only partially addressed:

- The introduction is better structured, but it still lacks explicit hypotheses. The objectives remain descriptive, and the study would benefit from clearer, testable hypotheses to frame the analyses.

- The discussion of adult versus sub-adult edge densities briefly acknowledges potential bias, but the treatment is thin. Differences in receiver density and detection range across habitats (e.g., dense seagrass vs. more open reef or shelf environments) could confound observed patterns, and this is not critically evaluated. More consideration of how array design and habitat-specific detectability may have influenced edge density outcomes would strengthen confidence in the ontogenetic differences reported.

- While the Results have been streamlined, some sections remain overly descriptive, with repetition of percentages and site names that could be condensed.

- The authors did not include details on handling duration. A brief summary of handling times would improve transparency.

**Do you want your identity to be public for this peer review?** For information about this choice, including consent withdrawal, please see our Privacy Policy

Reviewer #1: No

---

## [Author Response · Author response to Decision Letter 2]

13 Oct 2025

Reviewer #1: The authors have made a commendable effort to address the majority of my comments, and the manuscript is much improved in terms of organization, clarity, and methodological detail. However, I feel a few points remain only partially addressed:

- The introduction is better structured, but it still lacks explicit hypotheses. The objectives remain descriptive, and the study would benefit from clearer, testable hypotheses to frame the analyses.

Response: Hypothesis added.

- The discussion of adult versus sub-adult edge densities briefly acknowledges potential bias, but the treatment is thin. Differences in receiver density and detection range across habitats (e.g., dense seagrass vs. more open reef or shelf environments) could confound observed patterns, and this is not critically evaluated. More consideration of how array design and habitat-specific detectability may have influenced edge density outcomes would strengthen confidence in the ontogenetic differences reported.

Response: A new paragraph added discussing this.

- While the Results have been streamlined, some sections remain overly descriptive, with repetition of percentages and site names that could be condensed.

Response: Results more condensed.

- The authors did not include details on handling duration. A brief summary of handling times would improve transparency.

Response: Range of handling times added.

---

## [Editor Report · Decision Letter 2]

14 Oct 2025

Ontogenetic Shifts in Space Use and Habitat Selection of Tiger Sharks in The Bahamas

PONE-D-25-14869R2

Dear Dr. Ayres,

We’re pleased to inform you that your manuscript has been judged scientifically suitable for publication and will be formally accepted for publication once it meets all outstanding technical requirements.

Kind regards,

Joel Harrison Gayford

Academic Editor

PLOS ONE
---

## [Editor Report · Acceptance letter]

PONE-D-25-14869R2

PLOS ONE

Dear Dr. Ayres,

I'm pleased to inform you that your manuscript has been deemed suitable for publication in PLOS ONE. Congratulations! Your manuscript is now being handed over to our production team.

Kind regards,

on behalf of

Mr. Joel Harrison Gayford

Academic Editor

PLOS ONE